# Quantum Chemistry Study on the Structures and Electronic Properties of Bimetallic Ca_2_-Doped Magnesium Ca_2_Mg*_n_* (*n* = 1–15) Clusters

**DOI:** 10.3390/nano12101654

**Published:** 2022-05-12

**Authors:** Chenggang Li, Yingqi Cui, Hao Tian, Baozeng Ren, Qingyang Li, Yuanyuan Li, Hang Yang

**Affiliations:** 1Quantum Materials Research Center, College of Physics and Electronic Engineering, Zhengzhou Normal University, Zhengzhou 450044, China; zznu_lcg@163.com (C.L.); zznu_lf@163.com (H.T.); 2School of Chemical Engineering and Energy, Zhengzhou University, Zhengzhou 450001, China; zznurbz@163.com; 3School of Physics and Electronic Engineering, Sichuan University of Science & Engineering, Zigong 643000, China; zznmuszg@163.com (Q.L.); zznucyq@163.com (Y.L.); zznutyn@163.com (H.Y.)

**Keywords:** CALYPSO, DFT, stability, Ca_2_Mg*_n_* clusters

## Abstract

Here, by utilizing crystal structure analysis through the particle swarm optimization (CALYPSO) structural searching method with density functional theory (DFT), we investigate the systemic structures and electronic properties of Ca_2_Mg*_n_* (*n* = 1–15) clusters. Structural searches found that two Ca atoms prefer to occupy the external position of magnesium-doped systems at *n* = 2–14. Afterward, one Ca atom begins to move from the surface into the internal of the caged skeleton at *n* = 15. Calculations of the average binding energy, second-order difference of energies, and HOMO–LUMO gaps indicated that the pagoda construction Ca_2_Mg_8_ (as the magic cluster) has higher stability. In addition, the simulated IR and Raman spectra can provide theoretical guidance for future experimental and theoretical investigation. Last, further electronic properties were determined, including the charge transfer, density of states (DOS) and bonding characteristics. We hope that our work will provide theoretical and experimental guidance for developing magnesium-based nanomaterials in the future.

## 1. Introduction

Nanomaterials with small particle sizes, specific surface areas and high surface energies possess wide applications in chemistry, physics, biology, medicine, materials and nanodevices. Magnesium atoms contain *s*^2^ closed-shell electron configuration similarly to helium, which plays an essential role in aerospace, mobile electronics, automobile and biomedical applications [1,2,3,4]. Currently, metal-doped magnesium clusters with unique geometries and fascinating electronic properties have received considerable attention in magnesium-based multi-function materials.

In the past decades, many experimental techniques and theoretical studies have been reported for the related structures and properties of pure magnesium clusters [5,6,7,8,9,10,11,12,13,14,15,16,17,18,19]. For instance, on the experimental side, the transition points of Mg*_n_* clusters were determined at *n* = 20 by Diederich’s group [5]. By measuring the photoelectron spectra and observing the *sp* band gaps of Mg*_n_* (*n* = 3–35) clusters, Thomas et al. found that the anion magnesium clusters exhibited a metallic character from *n* = 18 [6]. On the theoretical side, the ground state van der Waals potential of a magnesium dimer was described by five essential parameters of the Tang–Toennies potential model [7].

Janecek et al. studied the structures of neutral and cationic Mg*_n_* (*n* to 30) clusters using the local spin density functional [8]. Based on the MP4 (SDTQ) and CCSD(T) levels, the electron affinities were calculated for magnesium dimers and trimers [9]. By using the spin-unrestricted density functional theory with a local density approximation, Gong and co-workers studied the electronic structures of Mg*_n_* (*n* < 57) clusters [10]. Their results found that more 3*p* electrons will be involved in the *sp* hybridization with the cluster size increasing. More recently, Akola and co-workers focused on the structural and electronic properties of Mg*_n_* (*n* < 13) clusters [11]. Their investigations showed that the onset of metallization of Mg*_n_* clusters is difficult to assign due to the energy gap and *sp* hybridization. 

Subsequently, electron binding energies, structural and electronic properties and the nonmetal-to-metal transition were studied systemically by Jellinek and Acioli [12,13]. In addition, by using ab initio theoretical methods (B3PW91, B3LYP and MP4), Lyalin et al. investigated the structure and electronic properties of neutral and singly charged Mg*_n_* (*n* = 2–21) clusters [14]. The above study by Lyalin et al. suggested that the hexagonal ring structure determines the cluster growth from Mg_15_. Moreover, the electronic shell effects and the Jellium-like behavior manifest themselves in the formation of geometrical properties; however, the shell effects do not determine the geometry of the Mg clusters completely. 

In addition, the energetic structural properties of neutral magnesium clusters Mg*_n_* (*n* = 2–22) were investigated utilizing density functional theory [15]. More recently, Heidari and Xia’s groups presented further details about the structural transition and electronic properties of Mg*_n_* (*n* = 10–56) and Mg*_n_*^0/−^ (*n* = 3–20) clusters, respectively [16,17]. Recently, Duanmu’s group investigated the geometries, binding energies, adiabatic ionization potentials and adiabatic electron affinities of Mg*_n_*^+/0/−^ (*n* = 1–7) clusters [18] and cohesive energies by using the CCSD(T) scheme with MP2/CBS correction for Mg*_n_* (*n* = 2–19 and 28), respectively [19].

Most recently, research found that heteroatom doping is an effective strategy to stabilize geometrical structures or to tune electronic properties. Up to now, based on the different quantum chemistry calculations, studies on metal-doped (Be, Al, Ge, Sn, 3*d* and 4*d* TM atoms) and nonmetal-doped (B, C, N, O, F and Si) magnesium clusters have harvested many great achievements [20,21,22,23,24,25,26,27,28,29,30,31,32]. More importantly, based on the CALYPSO structural searching method and DFT, the structures and electronic properties of Be-, Be_2_-, Sr_2_- and Ba_2_-atom-doped differently sized magnesium clusters have been systemically discussed by Zeng’s, Zhao’s and our groups [33,34,35,36,37]. For example, Zeng and co-workers found that BeMg_9_^0^, BeMg_9_^+1^ and BeMg_8_^−1^ clusters possess relative highly stability out of the studied systems [33].

Moreover, the stability mainly originating from the σ-type covalent bond, is formed by the interaction between Be-*s* and Mg-*p* orbitals. Zhao’s group performed the structural and electronic properties of BeMg*_n_* (*n* = 10–20) clusters and their anions [34]. The research concluded that the position of Be atom changes from completely encapsulated sites to surface sites after reverting to the caged magnesium motif. Subsequently, the structures and electronic properties have been investigated for two Be-, Sr- and Ba-atom-doped small-sized magnesium clusters by our group [35,36,37]. For Be_2_Mg*_n_* (*n* = 1–20) clusters, from *n* = 10, the structures transfer from 3*D* to filled cage-like frameworks [35].

Furthermore, in the small size, one Be atom prefers the surface sites, and the other Be atom tends to embed inside magnesium motif. However, for the large size clusters (*n* > 18), two Be atoms were completely encapsulated into magnesium cages. In addition, the Be_2_Mg_8_ cluster possesses robust stability, and Be-2*p* and Mg-3*p* orbitals revealed increasing metallic behavior. More interestingly, based on the same calculated method, studies on the structural evolution and electronic properties were performed for SrMg*_n_*^0/−^ (*n* = 1–12) clusters [36]. As a result, the tower-like framework of the Sr_2_Mg_8_ cluster possesses higher stability out of the studied systems.

Moreover, the stronger *sp* hybridization leads to stronger Sr-Mg bonds, which is supported by the analysis of the multi-center bonds. Subsequently, our groups systemically reported the structures and electronic properties of two-barium-atom-doped magnesium in both neutral and anionic species [37]. A pagoda-like Ba_2_Mg_8_ was determined by analyzing the relative stability.

Analysis of the molecular orbitals indicated that the high stability comes mainly from the interaction between Ba-6*s* and Mg-3*p* orbitals. In conclusion, the stronger *sp* hybridization leads to stronger M-Mg bonds and metallic behavior. Last, the geometric structures and electronic properties have been investigated systemically for lithium-doped magnesium clusters [38]. The results indicated that lithium atoms prefer to occupy the convex sites of LiMg*_n_* structures. The LiMg_9_ cluster possesses relatively higher stability. In addition, the charges transfer from the Li to Mg atoms, and there exists strong hybridization among *sp* orbitals.

Thus, metal-atom-doped magnesium clusters provide an effective approach to creating novel structures and electronic properties. As the same group of alkaline earth metals, Ca and Be, Mg, Sr and Ba have the identical valence electronic configuration of *ns*^2^. Nevertheless, due to the different electronegativity and atomic radius of Ca atoms compared with Be, Mg, Sr and Ba atoms, are there similar frameworks for two-calcium-atom-doped magnesium clusters? If yes, do these clusters possess novel electronic and bonding properties? How does hybridization change?

To date, minor investigations on the structures and electronic properties of two-calcium-atom-doped magnesium clusters have been reported in theoretical calculations and experimental works. Thus, in the present work, motivated by Be-, Mg-, Sr- and Ba-doped magnesium, we performed a systematic investigation for two-calcium-atom-doped magnesium Ca_2_Mg*_n_* (*n* = 1–15) clusters.

First, we conducted wide structural searching and precise structural optimization to explore the structural evolution rule. Second, determining the stable configuration of Ca_2_Mg*_n_* (*n* = 1–15) clusters was conducted by analyzing the stability properties. Finally, some electronic properties, such as the charge transfer, IR and Raman spectra, DOS and bonding characteristics, are discussed for Mg-doped alkaline-earth clusters. We hope that our investigations will provide a theoretical and experimental basis for studying the microscopic mechanism of magnesium doped with alkaline-earth nanomaterials.

## 2. Computational Detail

In the present section, a stochastic global search algorithm based on the CALYPSO structural searching method was used to obtain the lowest and low-lying isomers of Ca_2_Mg*_n_* (*n* = 1–15) clusters and Mg*_n_*_+2_ (*n* = 1–15) clusters [39,40,41]. Based on this method, the corresponding stable or metastable structures can be successful with the chemical composition and given external conditions [42,43,44,45,46,47,48,49,50], and the detailed searching process can be found in our reported papers [42,43,44,45,46,47]. The structural optimization and energy calculations employed the B3PW91 functional and 6−311+G(d) basis set for Ca and Mg atoms, respectively [51,52].

In particular, the B3PW91 functional has been widely tested for magnesium and magnesium-based clusters [35,36]. In addition, the spin multiplicity (1, 3, 5 and 7) is included, no imaginary frequencies are validated. All calculations were performed using the Gaussian09 program package [53]. In the following works, the relative stabilities of the ground state Ca_2_Mg*_n_* (*n* = 1–15) and Mg*_n_*_+2_ (*n* = 1–15) clusters were studied by computing the average binding energy (*E_b_*) and second-order difference energy ($\Delta_{2}E$). Subsequently, the IR and Raman spectra, DOS, molecular orbitals and AdNDP were systemically investigated using the Multiwfn software for the studied clusters [54]. To ensure the reliability of our computational method, the bond length (*r*_e_), vibration frequencies (*ω*_e_) and dissociation energies (*D*_e_) were calculated for Mg_2_, Ca_2_ and CaMg dimers, respectively.

For the Mg_2_ dimer, our calculated values were *r*_e_ = 3.651 Å, *D*_e_ = 0.0790 eV, which are in good agreement with the experimental results (3.891 Å, 0.0866 eV) [55]. For the Ca_2_ and CaMg dimers, there are no experimental values available. Our calculated results for the bond lengths, vibration frequencies and dissociation energies are 4.2667 Å, 72.32 cm^−^^1^ and 0.1478 V for the Ca_2_ dimer and 3.909 Å, 82.63 cm^−^^1^ and 0.1114 V for the CaMg dimer, respectively. Moreover, the bond length and frequency of Ca_2_ dimer are also in excellent agree with Soltani’s theoretical values (4.285 Å and 65.2 cm^−^^1^), respectively [56]. This indicated the reliability of the proposed method in this work.

## 3. Results and Discussions

### 3.1. Geometric Structures

In Figure 1, the lowest and lower-lying energy structures are present for Ca_2_Mg*_n_* (*n* = 1–15) and Mg*_n_*_+2_ (*n* = 1–15) clusters. These isomers with energies from low to high are designated by *n**a*, *n**b*, *n**c* and *n**d*. Moreover, the electronic states and point symmetry are also collected in Table 1. Simultaneously, the Cartesian coordinates of the lowest energy structures of Ca_2_Mg*_n_* (*n* = 1–15) clusters are given in Appendix A.

First, the lowest energy structures of the Mg*_n_* clusters agree with previous research by Zhang and Li et al. [35,37], which indicates that the present functional and basis sets are reliable. Moreover, the Ca_2_Mg*_n_* (*n* = 1–8) clusters possess similar geometric structures compared with the Mg*_n_*_+2_ clusters. Second, Ca_2_Mg possesses a triangular plane structure, and the lowest energy structures of Ca_2_Mg*_n_* (*n* = 2–15) clusters maintain a three-dimensional (3*D*) configuration. 

Specifically, Ca_2_Mg*_n_* (*n* = 2–6) clusters can be obtained by adding one Mg atom to the Ca_2_Mg*_n_*_−1_ clusters, and Ca_2_Mg*_n_* (*n* = 7–9) clusters can be generated from the substitution of Mg*_n_*_+2_ clusters by two Ca atoms, respectively. Interestingly, the lowest energy structure of the Ca_2_Mg_8_ cluster possesses the same geometrical form as those of X_2_Mg_8_ (X = Be, Sr and Ba) clusters. Finally, from 2 to 14, the doped two Ca atoms prefer to locate outside the host Mg*_n_*_+2_ cluster. At *n* = 15, one Ca atom starts to move from the surface into the internal of the caged skeleton. 

### 3.2. Relative Stability

To determine the relative stabilities of Ca_2_Mg*_n_* and Mg*_n_*_+2_ (*n* = 1–15) clusters, the average binding energy (*E_b_*) and second-order difference of energy $\Delta_{2}E$ are calculated as follows:(1)$$\begin{array}{l} E_{b}\left({\mathrm{Ca}}_{2}{\mathrm{Mg}}_{n}\right)=\frac{nE(\mathrm{Mg})+2E\left(\mathrm{Ca})-E({\mathrm{Ca}}_{2}{\mathrm{Mg}}_{n}\right)}{n+2}\\ E_{b}\left({\mathrm{Mg}}_{n}\right)=\frac{nE\left(\mathrm{Mg})-E({\mathrm{Mg}}_{n}\right)}{n}\\ \Delta_{2}\left({\mathrm{Ca}}_{2}{\mathrm{Mg}}_{n}\right)=E\left({\mathrm{Ca}}_{2}{\mathrm{Mg}}_{n-1}\right)+E\left({\mathrm{Ca}}_{2}{\mathrm{Mg}}_{n+1}\right)-2E\left({\mathrm{Ca}}_{2}{\mathrm{Mg}}_{n}\right)\\ \Delta_{2}E\left({\mathrm{Mg}}_{n}\right)=E\left({\mathrm{Mg}}_{n-1}\right)+E\left({\mathrm{Mg}}_{n+1}\right)-2E\left({\mathrm{Mg}}_{n}\right) \end{array}$$

*E* denotes the total energy of the corresponding clusters or atoms. The calculated results are plotted in Table 1 and Figure 2A,B. The following information can be concluded: (1) The values of *E_b_*(Mg*_n_*_+2_) clusters are lower than those of *E_b_*(Ca_2_Mg*_n_*) clusters, indicating that the doped Ca atom can improve the stability of pure magnesium clusters. (2) With an increase of Mg atoms, the values of *E_b_* (Mg*_n_*_+2_ and Ca_2_Mg*_n_*) monotonically increase, which indicates an enhanced effect on the stabilities for Ca_2_Mg*_n_* and Mg*_n_*_+2_ (*n* = 1–15) clusters. (3) $\Delta_{2}E$ values show a certain degree of oscillation, and there exist four visible peaks in the curves at *n* = 2, 5, 8, 11 and 13, implying that the Ca_2_Mg_2,5,8,11,13_ and Mg_4,7,10,13,15_ clusters possess higher relative stability. Most interestingly, the Ca_2_Mg_8_ cluster with 20 valence electrons possesses an notably high $\Delta_{2}E$ value (0.66 eV), which indicates superior relative stability.

The HOMO–LUMO energy gap (*E*_g_) is the other powerful tool to study the relative stabilities. Generally speaking, large values indicate stronger stability. In the present section, the HOMO–LUMO energy gaps are presented in Table 2 and Figure 2C. First, values of *E*_g_(Mg*_n_*_+2_) are larger than those of *E*_g_(Ca_2_Mg*_n_*) clusters, indicating that Ca_2_Mg*_n_* clusters are more stable, which is in reasonable agreement with previous research on averaged binding energies.

Second, Ca_2_Mg_2,6,8,9,15_ clusters with local maxima of *E*_gap_ suggests that those clusters are more stable than their neighbors. In summary, combining the conclusions of *E_b_*, Δ_2_*E* and *E_g_* values, the Ca_2_Mg_8_ cluster corresponds to the magic numbers and exhibits robust stability.

### 3.3. Charge Transfer

In this section, the charge-transfer information is analyzed by natural population analysis (NPA) in Table 1 and Figure 2D. Clearly, the doped Ca atoms possess positive charges in doped systems, meaning the charges transfer from calcium to magnesium atoms. 

Thus, the role of magnesium atoms is as charge acceptors, and Ca atoms are the charge donors. This is expected as Ca (1.00) has a smaller electronegativity compared with that of Mg (1.31) [57]. Second, the transferred charges increase with increasing cluster size. However, the value falls sharply at *n* = 15, and this situation may be the result of the position of Ca atom in the caged skeleton. In addition, the transfer charges in the range of *n* = 5–14 are all greater than 1.0 eV except for Ca_2_Mg_6_.

### 3.4. Infrared (IR) and Raman Spectra

In order to facilitate the characterization of spectra, we computationally simulated the infrared (IR) and Raman spectra of the lowest energy structures of Ca_2_Mg_8_ cluster. The simulated spectra with atomic labels are shown in Figure 3. The results found the highest intense IR frequency was located at 197.74 cm^−1^ with 4Mg-7Mg-10Mg bond tensile vibration; however, two Ca atoms were almost silent. The second and third strongest peaks can be found at 129.62 and 218.78 cm^−1^. For Raman spectra, the strongest peak at 181.51 cm^−1^ corresponds to the breathing vibration of all atoms. The second- and third-strongest Raman frequencies at 138.28 and 129.62 cm^−1^ correspond to the swing vibration of all atoms. In addition, IR and Raman spectra revealed that the strongest spectral frequencies are displayed in the range of 100–200 cm^−1^.

### 3.5. The Density of States

To understand the nature of the chemical bonding, the total density of states (TDOS) and partial density of states (PDOS) of the lowest energy structures of Ca_2_Mg*_n_* (*n* = 1–15) are displayed in Figure 4. The TDOS is represented by the khaki shade; PDOS of Ca-*s*, Ca-*p*, Mg-*s* and Mg-*p* AOs (atomic orbitals) are represented by the red and blue solid lines as well as magenta and green dotted lines, respectively. We found that the contribution to TDOS mainly comes from the PDOS of Ca-*s*, Mg-*s* and Mg-*p* AOs in the region of occupied orbitals.

This indicates that *sp* hybridization has occurred in Ca-Mg atoms and Mg-Mg atoms. In fact, the *sp* hybridization of Mg-Mg promotes the formation of Mg*_n_* frames in Ca_2_Mg*_n_* clusters. The *sp* hybridization of Ca-Mg can promote the interaction between the two doped calcium atoms and magnesium frames of Ca_2_Mg*_n_* clusters, which is also the main reason why the stability of Ca_2_Mg*_n_* is higher than that of their corresponding pure magnesium clusters.

### 3.6. Bonding Characters

Based on the above analyses, a pagoda-like Ca_2_Mg_8_ structure possesses superior stability. To illustrate the source of higher stability, the bonding nature, such as the MOs (molecular orbitals) and multi-center bonds, are discussed for the lowest energy structure of Ca_2_Mg_8_ cluster. The molecular energy levels and corresponding orbitals are presented in Figure 5. First, calcium and magnesium atoms are composed of the same valence configuration of *s*^2^, and Ca_2_Mg_8_ with 20 valence electrons meets the requirement of the Jellium model in terms of the valence electron number.

Moreover, the shell structures consist of one 1S orbital, three 1P orbital, five 1D orbitals and one 2S orbital, all of which are occupied by the paired electrons. The energy of 1S, 1P, 1D and 2S states are arranged in order from low to high without energy levels overlapping, and the energy levels are also relatively concentrated. In addition, all the splitting energy levels of 1D orbital are lower in energy than the 2S orbital. As a result, Ca_2_Mg_8_ cluster is a closed shell 1S^2^1P^6^1D^10^2S^2^ filled with 20 valence electrons.

In addition, the contributions of molecular orbital for Ca_2_Mg_8_ cluster were probed utilizing Multiwfn 3.8 program. The HOMO corresponding to 2S state involves Ca-*s* (30.98%), Mg-*s* (19.53%) and Mg-*p* (44.86%). The HOMO-*m* (*m* = 1–5) exhibit 1D state, in which HOMO-1 and HOMO-4 are mainly composed of Ca-*s* (10.62%, 26.60%), Mg-*s* (41.87%, 16.99% for) and Mg-*p* (43.70%, 53.33%), respectively, and the remaining 1D state orbital is mainly contributed to by Mg atoms. The compositions of HOMO-6, 7, 8 corresponding to three 1P orbitals comes mainly from the Mg-*s* and *p* AOs as well as to a small extent from the Ca-*s* and *p* AOs. In the case of HOMO-9 (1S), Mg atoms provide more than 90% of the orbital contribution. Hence, the *sp* hybridization between the Ca and Mg atoms could promote the interaction between the doped-Ca and host-Mg atoms and form stronger Ca-Mg bonds. 

To further elucidate the bonding patterns of the Ca_2_Mg_8_ cluster, adaptive natural density partitioning (AdNDP) is included, representing the bonding of a molecule in terms of *n*-center two-electron (*n*c-2e) bonds. Notably, 1c-2e and 2c-2e bonds mean a localized character; the *n*c-2e bond (*n* ≥ 3) belongs to the delocalized character. For the Ca_2_Mg_8_ cluster, ten delocalized bonds with different occupation number (ON) are shown in Figure 6, which corresponds to three delocalized 3c-2e σ-bonds, two delocalized 4c-2e σ-bonds, two delocalized 5c-2e π-bonds, two delocalized 7c-2e π-bonds and one delocalized 9c-2e π-bond, respectively. 

Three 3c-2e σ-bonds with an ON of 1.745–1.766 |e| are formed by two opposite trigonal 5Mg-8Mg-10Mg and 3Mg-6Mg-7Mg units and one 2Ca-5Mg-3Mg unit. Two 4c-2e σ-bonds are accountable for the bonding between the quadrangle 4Mg-6Mg-8Mg-9Mg units (ON = 1.89 |e|) and pyramid 1Ca-7Mg-9Mg-10Mg units (ON = 1.774 |e|), respectively. There are two 5c-2e π-type bonds with ON = 1.80 and 1.792 |e|, derived from two Ca-Mg units of 1Ca-2Ca-3Mg-5Mg-9Mg and 1Ca-3Mg-4Mg-5Mg-9Mg units, respectively. Two 7c-2e ON = 1.945, 1.903 |e| (two symmetric units) and one 9c-2e ON = 1.994 |e| exhibit π-type bonding character. In addition, to deeply understand the nature of bonding, the Wiberg bond orders are calculated and listed in Table 3. The results indicate that the strong Ca-Mg bonds are greater than Ca-Ca and most Mg-Mg bonds.

## 4. Conclusions

In summary, a detailed investigation of the structures and electronic properties of Ca_2_Mg*_n_* (*n* = 1–15) was performed using the CALYPSO searching method and DFT calculations. Structural searching found that two Ca atoms prefer to occupy the external position of magnesium-doped systems at *n* = 2–14 and that one Ca atom tends to move from the surface into the internal of the caged skeleton at *n* = 15. The size-dependent binding energies, second-order difference of energies, and HOMO–LUMO gaps found a pagoda-like Ca_2_Mg_8_ as a magic cluster that possessed higher stability. 

Upon charge transfer analysis, charges transferred from calcium to magnesium atoms. The simulated IR and Raman spectra of the magic cluster revealed that the strongest spectral frequencies were displayed in the range of 100~200 cm^−1^. In addition, the high stability of Ca_2_Mg_8_ with a 20 valence electron cluster possessed a closed-shell electron configuration of 1S^2^1P^6^1D^10^2S^2^ in terms of the Jellium model. Last, the *sp* hybridization of Ca-Mg and Mg-Mg bonds was confirmed by the molecular orbitals and AdNDP, which contribute to the high stability of the Ca_2_Mg_8_ cluster.

## Figures and Tables

**Figure 1 nanomaterials-12-01654-f001:**
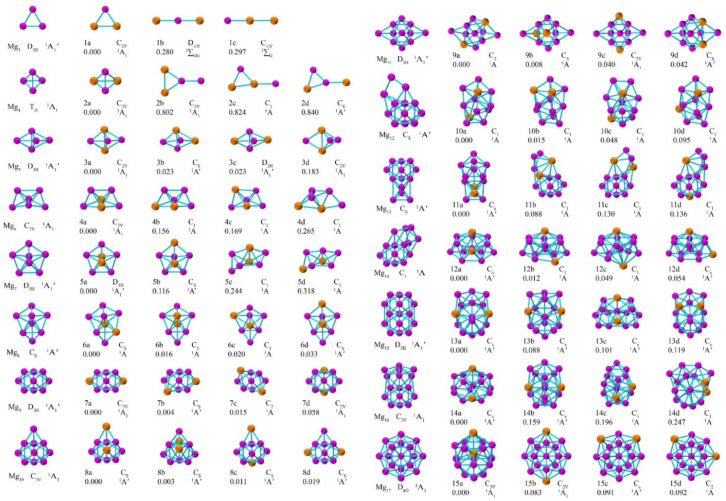
Optimized geometrical structures of lowest and low-lying isomers of Ca_2_Mg*_n_* (*n* = 1–15) clusters and Mg*_n_*_+2_ (*n* = 1–15) clusters along with the point group symmetry, electronic states and relative energy (eV). The pinkish and orange balls are magnesium and calcium atoms, respectively.

**Figure 2 nanomaterials-12-01654-f002:**
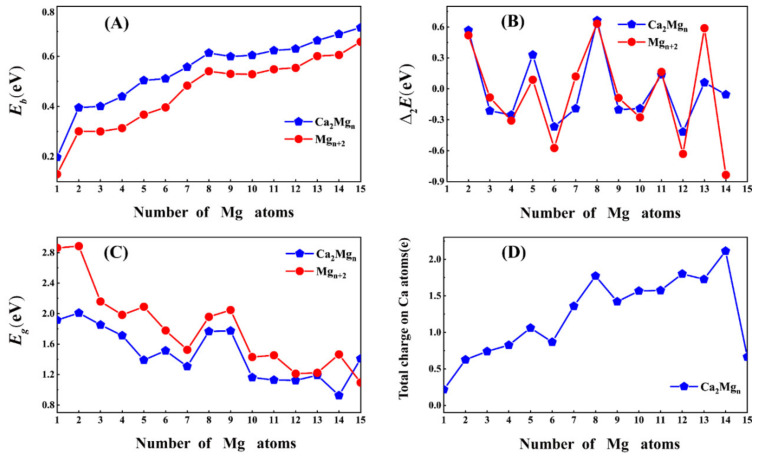
(**A**) *E_b_*, (**B**) Δ_2_*E*, (**C**) *E_g_* and (**D**) the total charges on Ca atoms in the ground state of Ca_2_Mg*_n_* (*n* = 1–15) clusters.

**Figure 3 nanomaterials-12-01654-f003:**
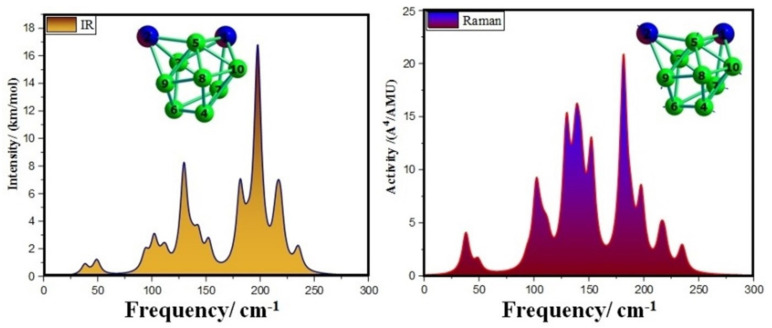
The Infrared and Raman spectra of the most stable cluster of the Ca_2_Mg_8_ cluster.

**Figure 4 nanomaterials-12-01654-f004:**
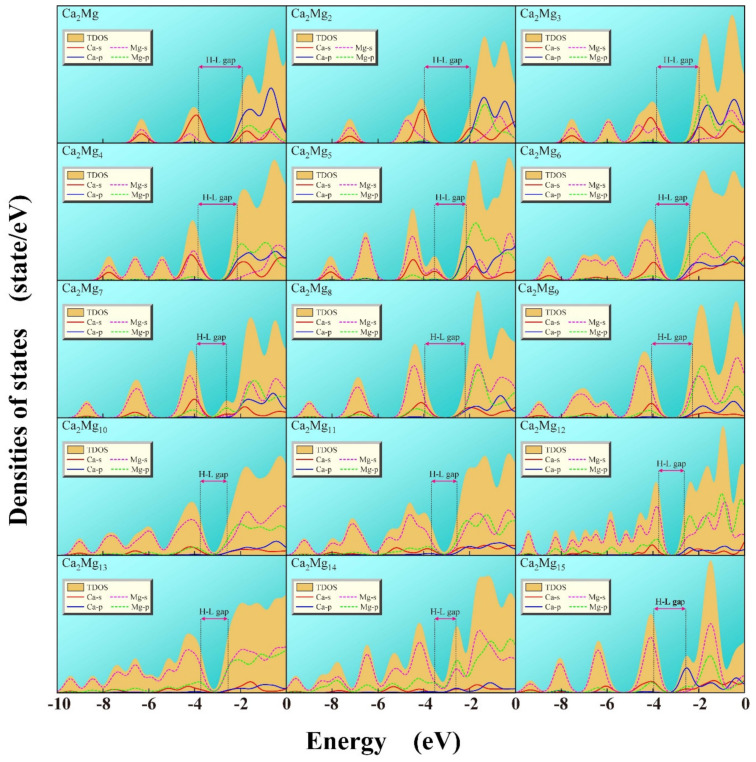
The calculated total densities of states (TDOS) and partial densities of states (PDOS) of Ca_2_Mg*_n_* (*n* = 1–15) clusters.

**Figure 5 nanomaterials-12-01654-f005:**
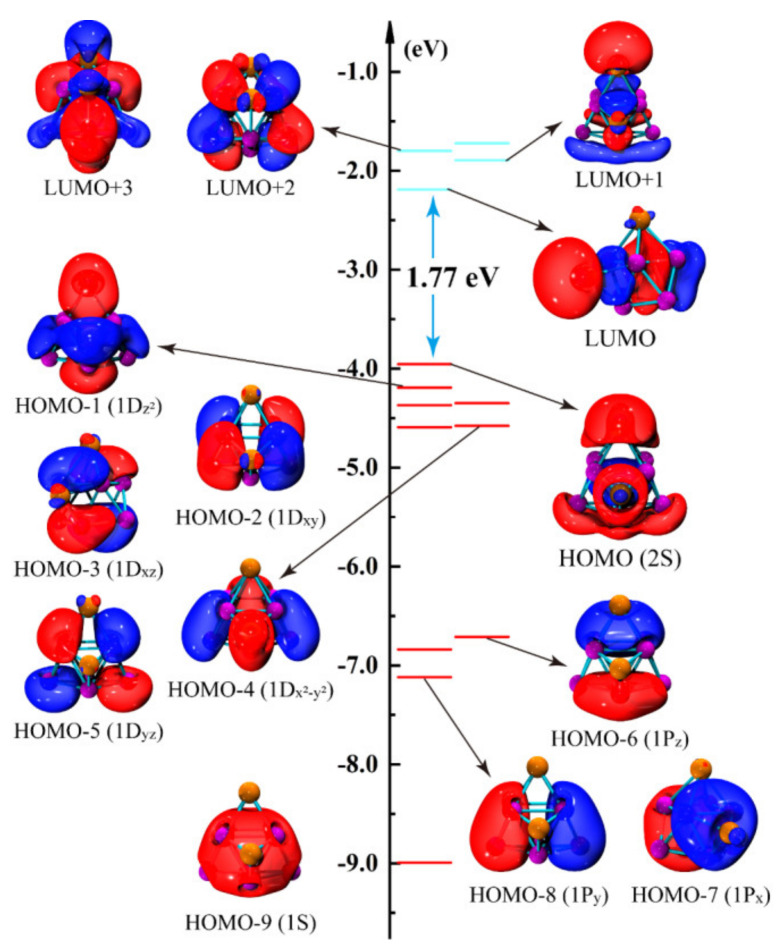
Molecular orbitals and the corresponding energy levels of the Ca_2_Mg_8_ cluster. The HOMO–LUMO gap is indicated (in azure).

**Figure 6 nanomaterials-12-01654-f006:**
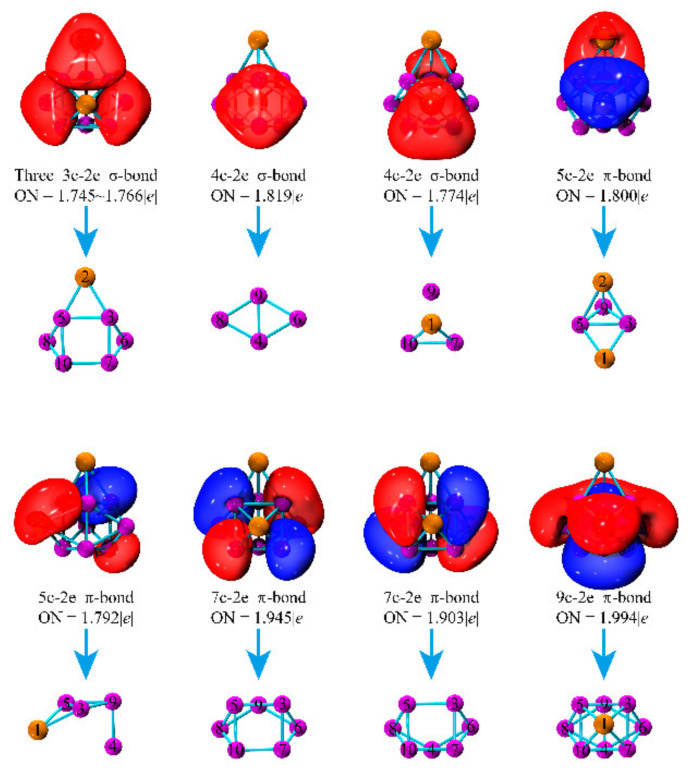
AdNDP chemical bonds and the corresponding structural units for the Ca_2_Mg_8_ cluster. (ON denotes the occupation number).

**Table 1 nanomaterials-12-01654-t001:** Electronic states, symmetries, average binding energies *E_b_* (eV), HOMO–LUMO energy gaps *E_g_* (eV), the second-order difference energy (∆_2_*E*) and charges on the Ca atoms of the most stable Ca_2_Mg*_n_* (*n* = 1–15) clusters.

Clusters	State	Sym.	*E_b_* (eV)	Δ_2_*E* (eV)	*E*_g_ (eV)	Charge (e)	
						Ca1	Ca2
Ca_2_Mg	^1^A_1_	C_2V_	0.20	-	1.91	0.11	0.11
Ca_2_Mg_2_	^1^A_1_	C_2V_	0.40	0.57	2.01	0.31	0.31
Ca_2_Mg_3_	^1^A_1_	C_2V_	0.40	−0.21	1.85	0.37	0.37
Ca_2_Mg_4_	^1^A_1_	C_2V_	0.44	−0.25	1.71	0.41	0.41
Ca_2_Mg_5_	^1^A_1_’	D_5H_	0.50	0.33	1.39	0.53	0.53
Ca_2_Mg_6_	^1^A	D_5H_	0.51	−0.37	1.51	0.37	0.37
Ca_2_Mg_7_	^1^A_1_	C_2V_	0.56	−0.19	1.31	0.68	0.68
Ca_2_Mg_8_	^1^A’	C_S_	0.61	0.66	1.77	0.84	0.94
Ca_2_Mg_9_	^1^A	C_2_	0.60	−0.20	1.77	0.71	0.71
Ca_2_Mg_10_	^1^A	C_1_	0.60	−0.19	1.16	0.86	0.71
Ca_2_Mg_11_	^1^A	C_S_	0.62	0.14	1.13	0.87	0.71
Ca_2_Mg_12_	^1^A’	C_S_	0.63	−0.42	1.12	0.85	0.85
Ca_2_Mg_13_	^1^A’	C_S_	0.66	0.06	1.19	0.86	0.86
Ca_2_Mg_14_	^1^A’	C_S_	0.69	−0.06	0.93	1.06	1.06
Ca_2_Mg_15_	^1^A_1_	C_3V_	0.71	-	1.41	−0.18	0.85

**Table 2 nanomaterials-12-01654-t002:** The HOMO and LUMO energy of Ca_2_Mg*_n_* and Mg*_n_*_+2_ (*n* = 1–15).

Clusters	HOMO (eV)	LUMO (eV)	Clusters	HOMO (eV)	LUMO (eV)
Ca_2_Mg	−3.834	−1.920	Mg_3_	−4.799	−1.937
Ca_2_Mg_2_	−3.981	−1.973	Mg_4_	−4.961	−2.076
Ca_2_Mg_3_	−3.831	−1.979	Mg_5_	−4.258	−2.099
Ca_2_Mg_4_	−3.860	−2.148	Mg_6_	−4.294	−2.311
Ca_2_Mg_5_	−3.531	−2.140	Mg_7_	−4.436	−2.347
Ca_2_Mg_6_	−3.897	−2.384	Mg_8_	−4.284	−2.507
Ca_2_Mg_7_	−3.917	−2.611	Mg_9_	−4.393	−2.868
Ca_2_Mg_8_	−3.954	−2.188	Mg_10_	−4.306	−2.350
Ca_2_Mg_9_	−4.048	−2.275	Mg_11_	−4.385	−2.338
Ca_2_Mg_10_	−3.750	−2.588	Mg_12_	−4.012	−2.583
Ca_2_Mg_11_	−3.671	−2.543	Mg_13_	−4.105	−2.652
Ca_2_Mg_12_	−3.746	−2.623	Mg_14_	−4.092	−2.881
Ca_2_Mg_13_	−3.730	−2.539	Mg_15_	−3.973	−2.750
Ca_2_Mg_14_	−3.518	−2.592	Mg_16_	−3.985	−2.521
Ca_2_Mg_15_	−3.965	−2.558	Mg_17_	−3.917	−2.823

**Table 3 nanomaterials-12-01654-t003:** The Wiberg bond orders of the Ca_2_Mg_8_ cluster.

Atoms	Ca-1	Ca-2	Mg-3	Mg-4	Mg-5	Mg-6	Mg-7	Mg-8	Mg-9
Ca-2	0.130								
Mg-3	0.383	0.462							
Mg-4	0.151	0.143	0.233						
Mg-5	0.383	0.462	0.524	0.233					
Mg-6	0.139	0.131	0.517	0.529	0.228				
Mg-7	0.398	0.149	0.528	0.439	0.237	0.523			
Mg-8	0.139	0.131	0.228	0.529	0.517	0.182	0.199		
Mg-9	0.179	0.410	0.465	0.458	0.465	0.487	0.231	0.487	
Mg-10	0.398	0.149	0.237	0.439	0.528	0.199	0.523	0.523	0.231

## Data Availability

The data are included in the main text.

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
