# Peer review of "Quantum Chemistry Study on the Structures and Electronic Properties of Bimetallic Ca2-Doped Magnesium Ca2Mgn (n = 1–15) Clusters"

_nanomaterials, 2022, doi:10.3390/nano12101654_

Round 1

Reviewer 1 Report

REVIEW FOR
Title: Quantum chemistry study on the structures and electronic properties of
two calcium-doped magnesium Ca2Mgn (n=1-15) clusters

Nano materials

TO AUTHORS:

This paper is well presented and a timely contribution to the literature. The English can be improved as indicated below. I would call the paper of average interest and originality  and still reasonable for publication.

  1. Title should be calcium 2 – doped ….
  2. p 1, line 4 from bottom use “due to” rather than “since”

The English can still use polishing but is nearly acceptable.

Only minor changes required prior to publication.

Author Response

Dear Editors and Reviewers:  
    Thank you for your letter and the reviewer's comments concerning our manuscript entitled “Quantum Chemistry Study on the Structures And Electronic Properties of Bimetallic Ca2-Doped Magnesium Ca2Mgn (n=1–15) Clusters” (ID: nanomaterials-1710117). Those comments are all valuable and very helpful for improving our paper. We have revised the whole manuscript carefully and tried to avoid any grammar or syntax error. Moreover, we have asked several colleagues who are skilled authors of English papers to check the English. And here we did not list the changes but marked in red in revised paper.

    We appreciate the Editors/Reviewers’ warm work earnestly, and hope that the corrections will meet with approval. Thank you very much again for your suggestions. We are looking forward to your information about my revised paper and thank you for your valuable suggestions.

  Yours sincerely,

Chenggang Li

Reviewer 1:

We are very appreciated for the Reviewer's comments and advice for our manuscript. We have revised carefully the whole manuscript. In addition, according to the other Reviewer’s suggestion, we have deleted some sentences and re-edited the whole manuscript. The track changes have been highlighted using red text in the revised manuscript. The responds to the reviewer’s comments are as following:

1. Response to comment: English language and style are fine/minor spell check required. Response: Thanks for your good advices. We have revised carefully the whole manuscript and tried to avoid any grammar or syntax error. We hope that English language is now acceptable for the next review process. 

  1. Response to comment: Are the results clearly presented?

Response: Thanks for your carefulness. Calculated results have been reedited and present with red text in our revised manuscript. 

  1. Response to comment: Title should be calcium 2 – doped ….

Response: Thank you very much for your suggestions. calcium 2 has been reedited using red text in our revised manuscript.  

  1. Response to comment: p 1, line 4 from bottom use “due to” rather than “since”

Response: Thank you very much for your suggestions. “due to” has been corrected in our revised manuscript.

Reviewer 2 Report

The applied calculation method and technical results look sound. However, the manuscript may be improved for general audience. The manuscript provides only minimal info as to what the novel properties of Ca2Mgn clusters are and why they are important in Abstract and Introduction. Various calculation results are presented, but many of them are not measurable.

It is recommended to include any experimental data that can validate the presented calculation results and implications/significance of the current findings for the general audience. 

Author Response

Dear Editors and Reviewers:  
    Thank you for your letter and the reviewer's comments concerning our manuscript entitled “Quantum Chemistry Study on the Structures And Electronic Properties of Bimetallic Ca2-Doped Magnesium Ca2Mgn (n=1–15) Clusters” (ID: nanomaterials-1710117). Those comments are all valuable and very helpful for improving our paper. We have revised the whole manuscript carefully and tried to avoid any grammar or syntax error. Moreover, we have asked several colleagues who are skilled authors of English papers to check the English. And here we did not list the changes but marked in red in revised paper.

    We appreciate the Editors/Reviewers’ warm work earnestly, and hope that the corrections will meet with approval. Thank you very much again for your suggestions. We are looking forward to your information about my revised paper and thank you for your valuable suggestions.

  Yours sincerely,

Chenggang Li

 Reviewer 2:

We are very appreciated for the Reviewer’s good comments for our manuscript. We have revised the whole manuscript carefully and tried to avoid any grammar or syntax error. In addition, according to the other Reviewer’s suggestion, we have re-edited the whole manuscript. The track changes have been highlighted using red text in the paper. The responds to the reviewer’s comments are as following:

1. Response to comment: English language and style are fine/minor spell check required. Response: Thanks for your good advices. We have revised the whole manuscript carefully and tried to avoid any grammar or syntax error. We hope that English language is now acceptable for the next review process. 2. Response to comment: Does the introduction provide sufficient background and include all relevant references?

Response: Thanks for your good advices. We are very sorry for our inadequacy description for background and references. We have added some references in our revised manuscript (see Introduction). Thank you very much again for your suggestions.

3. Response to comment: Are all the cited references relevant to the research?

Response: Thanks for your good advices. We have carefully read all the references in our revised manuscript. We believe that all the cited references are relevant to the research.

4. Response to comment: The manuscript provides only minimal info as to what the novel properties of Ca2Mgn clusters are and why they are important in Abstract and Introduction.

Response: Thanks for your good advices. We have added the some discussion on this issue in our revised manuscript. Thank you very much again for your suggestions.

5. Response to comment: Various calculation results are presented, but many of them are not measurable.

Response: Thanks for your good advices. As we all know, the special physical and chemical properties of cluster is closely linked with its microstructure. Therefore, it’s always the hot topic research in the field to determine the structure of clusters with the variation of the clusters. Currently, there still exist problems and challenges concerning the sole use of experimental measurement to determine the structure of a material, such as purity of the sample, quality of the signal and external conditions. Here, theoretical structure prediction is very important. It not only can assist experiment to determine the structures, but also can predict structure with novel physical and chemical properties, having profound guiding significance for experimental synthesis. So, in the present work, theoretical investigation on the structures and electronic properties are performed for Ca2Mgn (n=1–15) clusters. Various calculation results are obtained. However, it is important for general audience that experimental values are crucial. However, the structure and electronic properties of Ca2Mgn (n=1-15) clusters have been seldom reported unfortunately in theoretical and experimental aspects. So, we have added the some discussion about theoretical values compared with experimental results in Computational detail. We wish our theoretical investigation could provide more helpful for experimental investigation.

6. Response to comment: It is recommended to include any experimental data that can validate the presented calculation results and implications/significance of the current findings for the general audience.

Response: Thanks for your good advices. Indeed, some experimental data should be included. However, the structure and electronic properties of Ca2Mgn (n=1-15) clusters have been seldom reported unfortunately in theoretical and experimental aspects. So, in present work, some theoretical and experimental values are compared for Ca2, Mg2, and CaMg dimer. First, our theoretical results of the bond lengths (re), vibration frequencies (ωe), and dissociation energies (De) are present for Mg2 and Ca2 clusters. For Mg2 dimer, our calculated values are re=3.651Å, De=0.0790eV, which are in good agreement with the experimental results (3.891Å, 0.0866eV) [1]. For Ca2 and CaMg dimer, there no experimental values available. Our calculated results about bond lengths, vibration frequencies, and dissociation energies are 4.2667Å, 72.32cm-1 and 0.1478V for Ca2 dimer and 3.909Å, 82.63cm-1 and 0.1114V for CaMg dimer, respectively. Moreover, the bond length and frequency of Ca2 dimer are also in excellent agree with Soltani’s theoretical values (4.285Å and 65.2cm-1), respectively [2].

Our objectives in this work can be summarized as follows. First, the stable low-energy isomers of Ca2Mgn are searched and optimized by CALYPSO and DFT. This makes it possible to determine the ground-state structure by considering the total energy of the clusters together with their point group symmetry. Second, the size-dependent evolution of the ground-state structures of Ca2Mgn clusters can be expounded. Third, the electronic and bonding properties of Mg-doped alkaline-earth metals are discussed, to provide a theoretical basis for studying the microscopic mechanism of clusters doped with alkaline-earth nanomaterials.

Thank you very much again for your suggestions. We have added the some discussion on this issue in our revised manuscript (see Page 3).

Reference

  1. Ruette, F.; Sanchez, M.; Anez, R.; Bermudez, A.; Sierraalta, A. Diatomic molecule data for parametric methods.I. Mol. Struct: theochem 2005, 729, 19-37
  2. Soltani, A.; Boudjahem, A.G.; Bettahar, M. Electronic and magnetic properties of small RhnCa (n=1-9) clusters: a DFT Study. J. Quant. Chem. 2016, 116, 346-356
